# Free-Space Nonreciprocal Transmission Based on Nonlinear Coupled Fano Metasurfaces

**Ahmed Mekawy** [1,2] , **Dimitrios L. Sounas** [3] **and Andrea Alù** [1,2,4,*]

1. Department of Electrical Engineering, City College of The City University of New York, New York, NY 10031, USA; amekawy000@citymail.cuny.edu
2. Photonics Initiative, Advanced Science Research Center, City University of New York, New York, NY 10031, USA
3. Department of Electrical Engineering, Wayne State University, Detroit, MI 48202, USA; dsounas@wayne.edu
4. Physics Program, Graduate Center, City University of New York, New York, NY 10016, USA
* Correspondence: aalu@gc.cuny.edu

**Abstract:** Optical nonlinearities can enable unusual light–matter interactions, with functionalities that would be otherwise inaccessible relying only on linear phenomena. Recently, several studies have harnessed the role of optical nonlinearities to implement nonreciprocal optical devices that do not require an external bias breaking time-reversal symmetry. In this work, we explore the design of a metasurface embedding Kerr nonlinearities to break reciprocity for free-space propagation, requiring limited power levels. After deriving the general design principles, we demonstrate an all-dielectric flat metasurface made of coupled nonlinear Fano silicon resonant layers realizing large asymmetry in optical transmission at telecommunication frequencies. We show that the metrics of our design can go beyond the fundamental limitations on nonreciprocity for nonlinear optical devices based on a single resonance, as dictated by time-reversal symmetry considerations. Our work may shed light on the design of flat subwavelength free-space nonreciprocal metasurface switches for pulsed operation which are easy to fabricate, fully passive, and require low operation power. Our simulated devices demonstrate a transmission ratio >50 dB for oppositely propagating waves, an operational bandwidth exceeding 600 GHz, and an insertion loss of <0.04 dB.

**Keywords:** nonlinear metasurface; nonreciprocal metasurface; nonreciprocity; Fano resonance

## 1. Introduction

Optical nonreciprocity allows asymmetric light transmission from the opposite sides of a two-port device [1,2]. A common approach to break reciprocity is to utilize a magnetic bias [3,4], or in general a bias with odd symmetry under time-reversal, such that the propagating signals from opposite sides see entirely different devices. While a DC magnetic bias is bulky and difficult to integrate along with other components [3,4], the use of artificial angular or linear momentum bias achieved by periodically modulating the material parameters has enabled a paradigm shift to realize nonreciprocal components [5]. Time modulation has been successfully implemented on different platforms, from acoustics [6] to RF [7,8], THz [9], and optical frequencies [5,10],. Free-space nonreciprocal metasurfaces based on time modulation have been explored spanning the RF frequency range [11–13] and in the visible frequency range [14]. However, modulating the material parameters comes at the cost of power consumption, and it generates spurious harmonics that may interfere with nearby components [15], unless the devices are judiciously designed [16]. In addition, time modulation becomes harder as the operating frequency increases because the modulation frequency should scale accordingly [15]. Breaking the passivity of the system through the use of unidirectional gain elements (amplifiers) has also been explored to break reciprocity, leading to small size, ease of fabrication, and integrable CMOS technology for nonreciprocal responses [17–20]. However, the main disadvantage of such approaches is

the large power consumption and noise, because amplifiers are active elements that need to be properly biased in order to operate [20]. Additionally, the use of a CMOS amplifier limits the dynamic range of operation, so these metasurfaces are limited to work in the low power regime not exceeding a few dBm in order to avoid triggering nonlinearities in the amplifiers.

An alternative to these techniques, with the appealing property of not requiring any form of external bias or power, is the use of nonlinearities [21] combined with geometrical asymmetries. Light waves traveling through these devices can modulate the material refractive index while passing through it [22], and the geometrical asymmetry ensures that the induced variations are different for opposite propagation directions, breaking reciprocity. Due to the typically weak optical nonlinearity of nonlinear materials, these self-biased nonreciprocal devices have to rely on strong optical resonances and large contrast in the induced field distributions [23–25].

An ideal two-port optical isolator supports unitary transmission in one direction and full absorption (zero transmission) in the opposite direction, where the deviation from unitary transmission in the forward direction defines the insertion loss and the finite transmission in the reverse direction defines the isolation of the device [26]. In reality, nonlinearity-based nonreciprocal devices do not work like conventional isolators, in that they do not absorb the backward power, but instead reflect it [23,25,27]. While it is strictly forbidden by thermodynamic considerations to realize a passive two-port isolator using a lossless material, as we consider throughout our work [28,29], we emphasize that nonlinearity-based nonreciprocal devices are not true isolators, as they cannot isolate a weak backward-propagating signal in the presence of a strong forward signal [29], as dictated by dynamic reciprocity [30]. Therefore, our nonreciprocal device only ensures asymmetric transmission features in a switched mode operation, based on which it is separately excited at the two ports. Thermodynamic considerations [31–33] also imply that asymmetric transmission cannot arise for all power levels, as is consistent with the fact that nonreciprocity is inherently rooted in the nonlinear response of the system.

Different metrics can be used to characterize the performance of the devices under consideration. We can define a Nonreciprocal Intensity Range (NRIR) [5,27] as the range of input intensities for which we expect to find a largely different transmission level for opposite excitations. Interestingly, a nonlinearity-based nonreciprocal device based on a single nonlinear resonator, with an arbitrary Fano lineshape, suffers from an important trade-off between the minimum insertion loss in the system and its NRIR [27]. These limitations, along with the high power requirements and the associated signal distortions, hinder several applications of these concepts, despite their appealing bias-free, fully passive nature. Only recently, a few proposals to realize free-space nonreciprocal devices based on these concepts have been presented, based on coupled dielectric spheres in multilayers [23] or narrow slits in a dielectric grating [24,27], in addition to proposals based on guided wave implementations in optical [1] and RF frequencies [34]. In this work, we show theoretically and with full-wave simulations that, by using nonlinear metasurface bilayers separated by a subwavelength spacer, we can obtain a nonreciprocal compact device that overcomes the trade-off between insertion loss and NRIR in a free-space implementation. Moreover, we discuss the design guidelines and the response of the device at different wavelengths in order to assess its spectral bandwidth.

## 2. Principle of Operation of Nonlinearity-Based Nonreciprocal Devices Based on Coupled Fano Metasurfaces

Figure 1a schematically depicts the functionality of the nonlinear metasurface bilayer described in this work; a few designs will be explored in the following sections to demonstrate both high and low power operations. We use silicon (Si) as the material of choice, given its wide availability, compatibility with CMOS fabrication processes, and relatively large third-order nonlinear susceptibility $\chi^{(3)} \approx 2.8 \times 10^{-18}$ m$^2$/V$^2$ [22]. The choice of Si is also motivated by the possibility of comparing our results with recent reports using silicon metasurfaces for similar purposes [24,27]. We use glass as the spacer in our design due

to similar reasons, apart from the fact that it has no appreciable optical nonlinearity in the frequency and intensity ranges of interest. The device operation in Figure 1a can be explained in analogy with the microwave circuits introduced in [34], but with the Fano resonance realized using dielectric metasurfaces instead of electrical circuits. Fano resonant metasurfaces have been widely studied in nanophotonics. More recently, there has been significant work devoted to realize high-quality factor Fano resonances for optical and visible frequencies based on bound states in the continuum (BICs) [35–41]. Due to their enhanced light–matter interactions, these BIC metasurfaces can be efficiently employed in nonlinear optics applications [35].

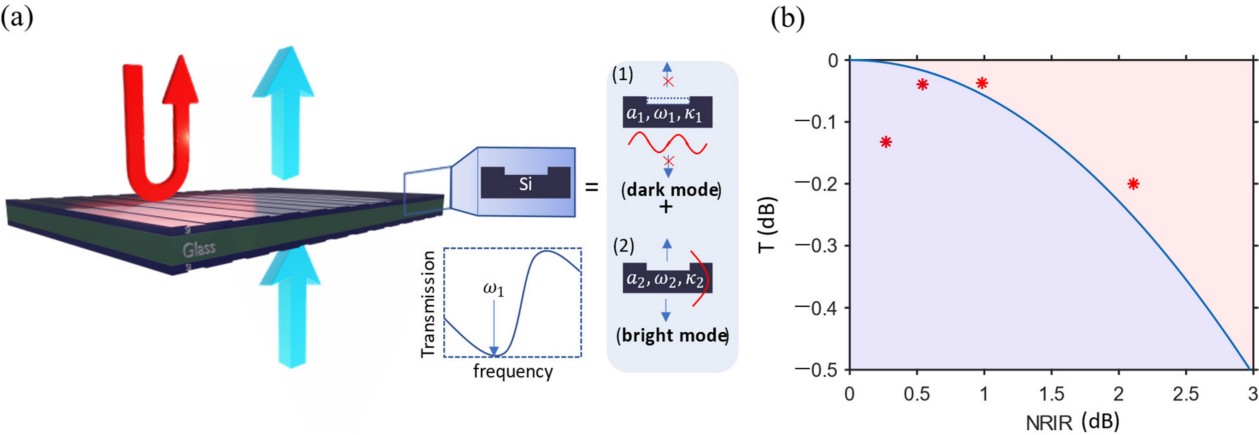

**Figure 1.** (**a**) Schematic depicts the nonlinearity-based nonreciprocal metasurface bilayer presented in this work. The arrows represent high power plane waves pointing to the direction of propagation that are blocked and reflected when propagating from the top to the bottom (red arrow) and transmitted when propagating from the bottom to the top (blue arrows). The zoomed in box shows one unit cell of the metasurface top layer made of Si (the bottom layer is similar, but with different geometrical parameters). Upon the plane wave excitation, the transmission from each layer takes the Fano line shape in the dashed box, and can be explained by the interference between: (1) the dark narrow band mode similar to the waveguide mode inside the unperturbed slab (i.e., with filled Si in the dashed box) and (2) the bright wideband mode that resembles a Fabry–Pérot mode or background reflection, as shown in the top and bottom of the shadowed rectangle, respectively. ($\omega_1$, $\omega_2$) are the resonance frequencies of the modes, while their complex amplitudes are ($a_1$, $a_2$) and their coupling rates to the free space are ($\kappa_1$, $\kappa_2$) for the dark and bright mode, respectively. (**b**) Transmission in the forward direction vs the nonreciprocal intensity range for the various nonlinear resonator designs presented in this work (red asterisks). The blue shaded region corresponds to the bound in Equation (5).

To start, we designed the top and bottom silicon surfaces in Figure 1a such that each layer supports a Fano resonance in the frequency range of interest based on the guided mode resonances concept (GMR) [24]. The zoomed-in box in Figure 1a shows one unit cell of the top metasurface layer. The Fano resonance of each layer can be explained similarly; here, we focus only on the top layer, of which the resonance can be explained by considering the interference between the broadband (bright) and narrowband (dark) modes. The (dark, bright) mode has a complex amplitude ($a_1$, $a_2$), resonance frequency ($\omega_1$, $\omega_2$) and linewidth ($\kappa_1$, $\kappa_2$) defining its coupling to free space. Here, these two modes correspond to (1) the waveguide mode that can be coupled to free-space radiation through the periodic perturbation along the slab interface, and (2) the Fabry–Pérot mode of the Si slab, or the broadband reflection from this slab, as shown in the top right and bottom right panels of Figure 1a, respectively. In general, $\kappa_2 \gg \kappa_1$ because the bright mode has a much larger coupling rate to the radiation continuum than the dark mode has. Therefore, when excited by plane waves, these modes interfere, creating a Fano line shape in the transmission curve due to their different coupling rates, as shown in the dashed box in Figure 1a. We define $\omega_0$ as the resonance frequency of the Fano resonator, at which the linear transmission coefficient goes to zero. This frequency also equals the resonance frequency of the dark mode, $\omega_0 = \omega_1$, as shown in the dashed box in Figure 1a. In particular, when we

excite the Fano resonator with an incident plane wave with frequency $\omega$, the transmission curve takes the form (see Equation (8) in Section 5.1 (Coupled Mode Theory))

$$T = \frac{1}{1+x^2} ; x = \left( \frac{\kappa_1}{\delta_1} + \frac{\kappa_2}{\delta_2} \right), \tag{1}$$

where $\delta_1 = \omega - \omega_1$ and $\delta_2 = \omega - \omega_2$. It is readily seen that, when $\delta_1$ or $\delta_2 = 0$, we have zero transmission; therefore, $\omega_1 = \omega_0$ in a linear case, and when $\kappa_1 \delta_2 = -\kappa_2 \delta_1$ we have unitary transmission. Additionally, because we assume $\kappa_2 \gg \kappa_1$ because it describes the bright mode, it follows that the mode frequency $\omega_2 \gg \omega_1$ to obtain unitary transmission is very close to the zero transmission and we have appreciable Fano resonance, as shown in the dashed box in Figure 1a.

Next, we now include the Kerr nonlinearity of Si, so that the refractive index $n$ changes as a function of the incident field intensity according to the relation $n = n_0 + n_2 I$, where $n_0$ is the refractive index at low incident power and $n_2$ characterizes the Kerr nonlinearity according to the relation $n_2 = 3\sqrt{\mu_0/\epsilon_0} \Re\left\{\chi^{(3)}\right\}/\left(4n_0^2\right) > 0$. When we have high incident power, the resonance frequency $\omega_1$ is no longer constant, i.e., $\omega_1 \neq \omega_0$; instead, it depends on the frequency and power of the incident wave, such that (see Equation (11) in Section 5.2 (Nonlinear Bistability))

$$\delta_1 = \delta_{01} + \frac{\left(\frac{\kappa_1}{\delta_1}\right)^2}{1 + \left(\frac{\kappa_1}{\delta_1} + \frac{\kappa_2}{\delta_2}\right)^2} P, \tag{2}$$

where $\delta_{01} = \omega - \omega_0$ is the frequency shift where we recall that $\omega_0$ is the resonance frequency of mode 1 before including the nonlinearity, and $P$ is a normalized term proportional to the incident power. Because we assume that the mode frequency $\omega_2$ is very far away from the frequency range of interest, its properties are weakly affected by the high power, so we assume its parameters $\omega_2$ and $\kappa_2$ to be fixed in the nonlinear analysis. In the following, we study optical switching in nonlinear Fano resonators near their resonance frequencies as a function of the incident power $P$ and input frequency $\omega$.

First, we assume an incident monochromatic wave with fixed normalized power ($P = 0.5$), and we study the response of the nonlinear Fano resonance as a function of the input frequency $\omega$. Inspecting Equation (2), we can immediately recognize that a regime can arise for which the response of the nonlinear Fano resonator becomes bistable, a scenario of particular importance for the findings described in this paper. In particular, we expect a bistable response at some threshold input power if (see Section 5.3 Bistability Condition))

$$\frac{\omega_{\text{mod}} - \omega}{\kappa_{\text{mod}}} > \sqrt{3}, \tag{3}$$

where $\omega_{\text{mod}} = \omega_1 - \frac{\Delta_\omega \kappa_1 \kappa_2}{\Delta_\omega^2 + \kappa_2^2}$ and $\kappa_{\text{mod}} = \frac{\Delta_\omega^2 \kappa_1}{\Delta_\omega^2 + \kappa_2^2}$. In Figure 2a, we show the resonance frequency shift $\delta_1$ as a function of the applied incident frequency for $P = 0.5$. Generally, the resonance frequency shift increases with the incident frequency; however, in a range of frequencies slightly below the resonance frequency for low intensities $\omega_0$, it abruptly jumps to the upper branch of the bistability curve, as indicated by the black arrow at the critical point $c_1$. The further increase of the incident frequency leads to a monotonic increase of the resonance shift. If we now consider a decreasing input frequency, we find an abrupt jump of the resonance shift from the upper branch to the lower branch at the critical point $c_2$, implying a hysteresis in the bistable response. Interestingly, for Fano resonators, we can choose the parameters ($\kappa_1$, $\kappa_2$, $\omega_1$, $\omega_2$) such that this discontinuous jump enables zero to unitary switching in transmission, by requiring $\delta_1 = 0$ at the lower branch, while $\kappa_2 \delta_2 = -\kappa_1 \delta_1$ at the upper branch. This particular scenario is identified by the horizontal dashed line in Figure 2a, where we choose the parameters to ensure $\delta_1 = 0$ at the point $c_1$

on the lower branch, and $\delta_1\kappa_2 = -\delta_2\kappa_1$ on the upper branch. This interesting response may find potential applications for frequency filters [22]. In order to demonstrate the abrupt zero to the unitary transmission feature, we plot the transmission of this Fano resonator as a function of the incident frequency in Figure 2b, where it indeed shows that the point $c_1$ offers very low transmission and, as we increase the frequency beyond a threshold value, the transmission abruptly switches to unity (0 dB). In general, we can design the bistability region as desired, e.g., to have the abrupt change in transmission at some desired frequency; however, we emphasize that, because we assumed $n_2 > 0$, the bistability can only happen for input frequencies smaller than the resonance frequency at low intensities. We can prove this important property by solving Equation (2) exactly such that $\delta_1$ has a single real solution, or we can graphically compare the bistable transmission curve to the linear transmission curve, i.e., the solid blue line in Figure 2b, showing that we can find a bistability only when the incident frequency is lower than the linear resonance frequency, as in the case of nonlinear Lorentzian resonators [42]. This behavior can be observed for any desired power level $P$, as shown in Figure 2c. A larger incident power generally results in a lower switch-up frequency threshold and a wider hysteresis of the bistable response, and in the limit of very low power levels the transmission converges to the linear scenario, confirming again that the bistability can only occur for incident frequencies lower than the resonance frequency. The opposite remarks apply for $n_2 < 0$. Figure 2c confirms that, for $P = 0.5$, we find an abrupt change in transmission from 0 to 1, corresponding to the dashed line in Figure 2a.

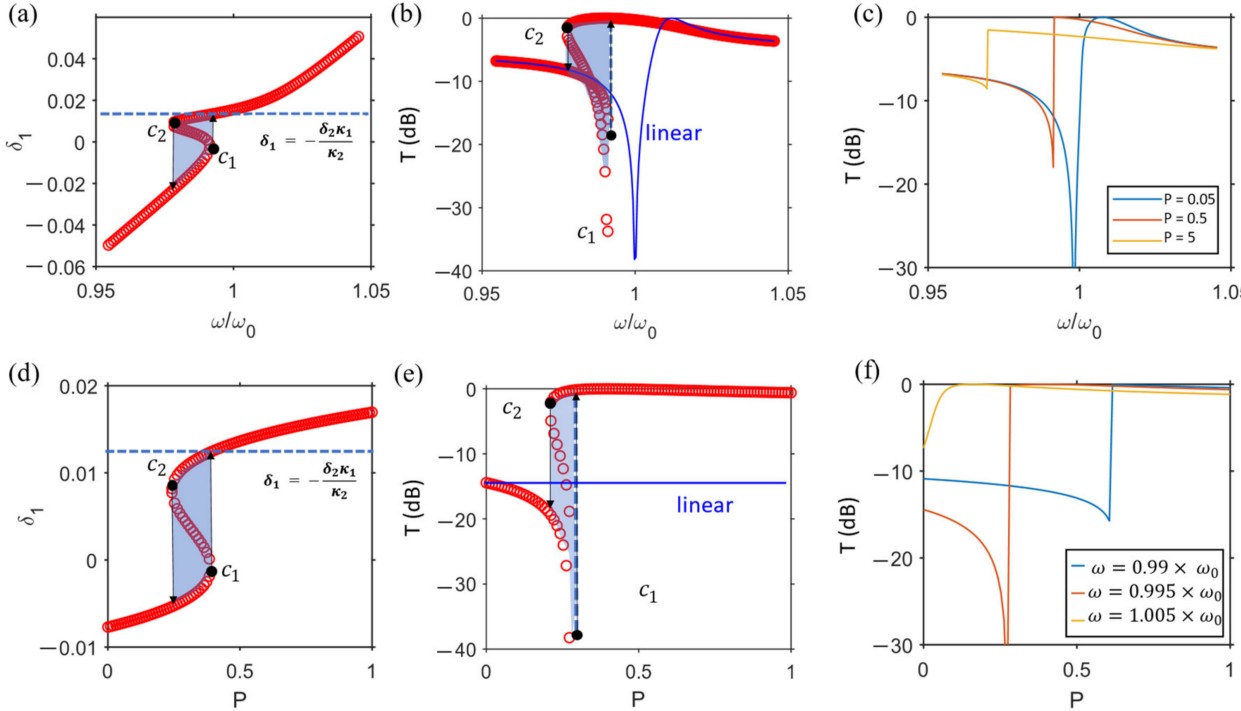

**Figure 2.** Nonlinear response of individual Fano resonators as a function of the input frequency and power. (**a**) Bistability in the shift of the zero-transmission frequency, $\delta_1$, versus the frequency $\omega$ of the input plane wave, with fixed input power $P = 0.5$. For an incident wave with an increasing frequency, the shift follows the path through the point $c_1$ along the arrow direction to the top branch, and for a decreasing frequency the shift path follows the line passing through the point $c_2$ along the arrow direction to the lower branch. The dashed lines indicate the value $\delta_1$, such that the transmission coefficient goes to 0 dB. (**b**) Similar to (**a**), but with the transmission coefficient instead of shift $\delta_1$. The blue line plots the linear response of the resonator, i.e., at $P = 0$. (**c**) Similar to (**b**) but considering different power levels and only branches with an increasing frequency. (**d**–**f**) Similar to (**a**–**c**) but analyzing the variations with the input power at fixed frequency $\omega = 0.995 \times \omega_0$. The other parameters are constant for all of the subfigures, $\omega_0 = 1.1$, $\kappa_2 = 152$, $\omega_2 = 100$, and $\kappa_1 = 0.02$.

Next, we study the nonlinear Fano resonator properties for fixed incident frequency while changing the input power $P$. Similar conclusions can be drawn by observing the results in Figure 2d–f. For instance, Figure 2d shows the nonlinear shift $\delta_1$ as a function of the incident power $P$ while keeping the frequency fixed ($\omega = 0.995\omega_0$). Similar to Figure 2a, we select the parameters that enable an abrupt jump in transmission from 0 ($-40$ dB) to 1 (0 dB) as we increase the incident power. This can be confirmed in the transmission plot in Figure 2e, in which at the critical point $c_1$ the lower branch shows 0 transmission, and a slight increase in power causes an abrupt jump to unitary transmission. This behavior has potential in optical applications like optical limiters, nanoswitches [22,43] and nonreciprocal devices [23]. Similar to Figure 2c, we plot in Figure 2f the transmission as the input power increases for different frequencies. We observe again that the bistability only arises when the input frequency is lower than the resonance frequency of the linear resonator. In addition, as the incident frequency becomes smaller, the power needed to trigger a bistable transition becomes larger. Because this bistability regime can be used to induce a nonreciprocal response, by making sure that the system operates in different stable states for opposite propagation directions, we deduce that a low-power nonreciprocal device can be achieved if the operating frequency is close to, and less than, the resonance frequency of the linear Fano resonator (Figure 2c).

Having analyzed the response of a single Si layer operated as a Fano resonator, we can now consider nonlinear optical devices based on coupled nonlinear Fano resonators, as shown in Figure 1a, where each Si layer supports a Fano resonance with its own resonance frequency and decay rate, and they are separated from each other by an electrical length $\theta$. A functional working principle of a nonreciprocal device based on this geometry consists in the following scheme: for a given excitation frequency and power, one of the resonators (resonator 2) is designed to enter its bistable region, while the other one (resonator 1) remains outside this region. In order to distinguish the dark mode resonance frequency of each Fano resonator, we call the dark mode of resonator 1 $\omega_{01}$, while the dark mode of resonator 2 is $\omega_{02}$, and we recall that $\omega_{0i}$ is the frequency at which the linear transmission goes to zero. Therefore, we can design resonator 2 with the linear resonance frequency $\omega_{02} > \omega_{in} > \omega_{01}$. The linear spectral transmission curves for resonator 1 and resonator 2 are shown in Figure 3a for the parameters in Table 3 in the first and second rows in Section 5.5 (Fano Resonator Parameters) for resonator 1 and resonator 2, respectively. We now consider an incident excitation with increasing power $P$ and of frequency $\omega_{in}$ indicated by the vertical dashed line in Figure 3a, such that $\omega_{01} < \omega_{in} < \omega_{02}$. As we increase the power of the incident wave, resonator 2 exhibits a sharp transition in the transmission curve at the critical point $P = 0.7$, confirming that it operates in the bistable region, while resonator 1 maintains a smooth variation of transmission, as shown in Figure 3b. For operation as an ideal nonreciprocal device, it is particularly important to design the two resonators to have unitary transmission at the same power level [31], as shown in Figure 3b for the intensity around $P = 1$. By placing the resonators separated by an electrical distance $\theta$, as shown in the inset of Figure 3c, we can test the working principle of this device by exploring the response for excitation from opposite sides. First, we excite the structure with frequency $\omega_{in}$ from the side of resonator 2. Initially, with the very small incident power $P$, the reflection is large and, as we increase the power, we reach the critical point at which the transmission for resonator 2 goes from 0 to 1; therefore, the signals are transmitted from resonator 2 to resonator 1. Because we designed resonator 1 to support unitary transmission at the same critical power as resonator 2, the incident signal is transmitted to the left port. Overall, the transmission as a function of intensity for excitation from right to left $T_{RL}$ is similar to the transmission of the individual resonator 2, as shown by the dashed line in Figure 3c. Next, we consider the excitation from the left side. Again, we start with a small input power, for which the incident wave is partially transmitted to resonator 2. Because at this power level resonator 2 is highly reflective, the wave experiences multiple

reflections between the resonators, and it reaches a steady-state transmission level $T_a$. The incident power on resonator 2, $P_{\text{eff}}$, can be calculated as (see Equation (18) in Section 5.4)

$$P_{\text{eff}} = \frac{T_a}{2 - T_a - 2\sqrt{1 - T_a}\cos(\phi_a + \phi_b + 2\theta)} P, \tag{4}$$

where $T_a$ is the nonlinear transmission for excitation from the side of resonator 1, and $\phi_a$ and $\phi_b$ are the nonlinear reflection phases from resonator 1 and resonator 2, respectively, as shown in Figure 3e, which are not to be confused with the reflection phase in the linear regime shown in Figure 3d. While the former shows an abrupt change in phase at the bistability transition, the latter obviously does not.

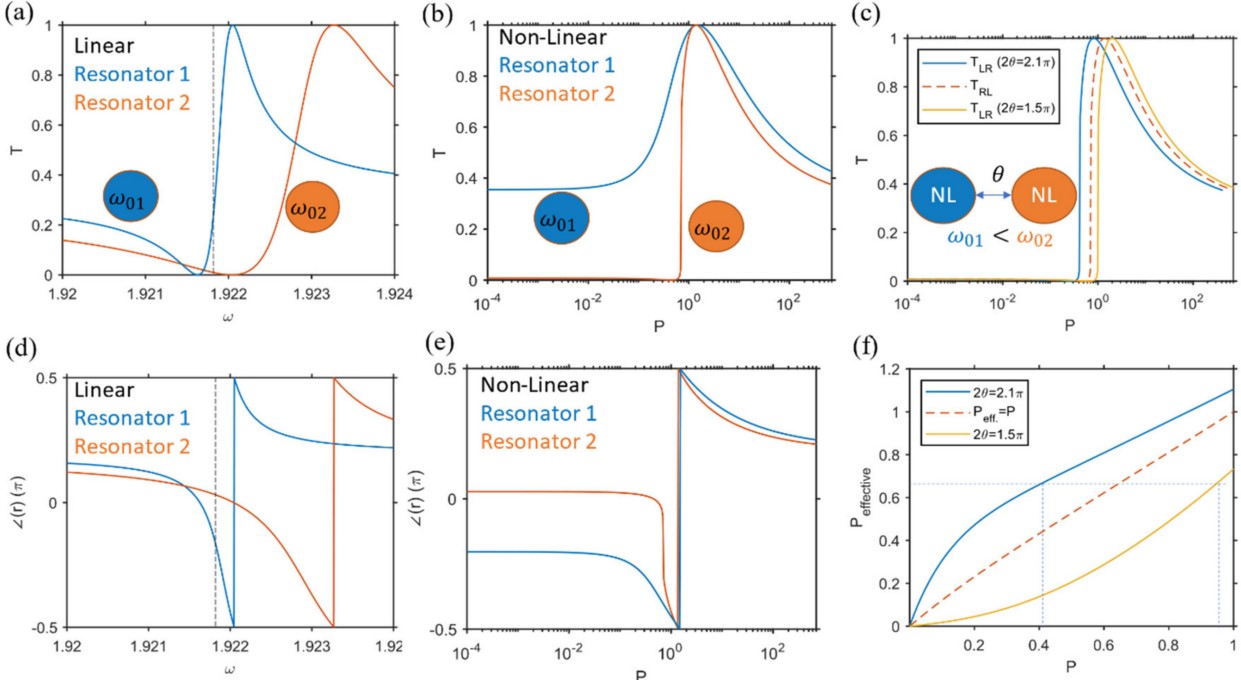

**Figure 3.** Operation of the device based on coupled shifted Fano resonators. (**a**) Linear transmission coefficient for two individual resonators with different zero transmission frequencies. $\omega_{01}$ is the zero-transmission frequency of the first resonator, and $\omega_{02}$ is for the second resonator. (**b**) Nonlinear transmission coefficient of the resonators in (**a**) excited at the frequency dictated by dashed lines in (**a**) with increased power $P$. (**c**) Nonlinear transmission coefficient for coupled resonators, separated by a delay line $\theta$ with increasing $P$. The transmission from left to right is $T_{LR}$, while that from righ to left is denoted as $T_{LR}$. (**d**,**e**) Linear and nonlinear reflection phases of the individual resonators. (**f**) The effective power seen by the second resonator when the device is excited from left for different electrical lengths $\theta$. The vertical dashed lines indicate the required input power for the transition to happen in the transmission curve for the corresponding thetas. We used the parameters for resonator 1 and resonator 2 as given in Table 3 (row 1 and row 2) in Section 5.5 (Fano Resonator Parameters).

The solution of Equation (4) considers an effective value of $T_a$ stemming from the multiple reflections between the two resonators, which may be calculated based on numerical simulations, but it can also be evaluated with good approximation assuming that $T_a$ is only affected by the incident power from the left, ignoring the reflected power from resonator 2. In other words, when the wave experiences the first round trip, resonator 1 will see two incoming waves: the incident wave from left, and the reflected wave from resonator 2 from the right. Therefore, $T_a$ should be updated based on the total input power from left and right. This process should repeat; in each iteration we update $T_a$ based on the total incoming power from left and right until we reach a steady state; however, in Equation (4) we assumed that $T_a$ is updated only once, neglecting that the reflection from resonator 2 will update the transmission $T_a$. This approximation makes it easier to develop physical

insights into the response of the system; however, as shown in the numerical results in the next section, this approximation is quite accurate. We calculate the value of $P_{eff}$ for different electrical lengths $\theta$ as a function of the incident power $P$, as shown in Figure 3f. It is evident that $P_{eff}$ can be larger or smaller than $P$, thereby shifting the transmission curve of resonator 2 to higher or lower input power, respectively. In particular, for $2\theta = 2.1\pi$, $P$ should be $0.4 < 0.7$ in order to obtain a bistable transition in resonator 2 which is lower than the transition power when excited from the right; in contrast, $2\theta = 1.5\pi$ shows that $P$ should be $0.95 > 0.7$ in order to enable the bistable transition. These particular values are indicated by the vertical dashed lines in Figure 3f. For the two values of $\theta$ considered, we plot the transmission from left to right $T_{LR}$, as shown in Figure 3c. Thus, we obtain a nonreciprocal response with a large transmission from left to right, and zero transmission from right to left for a specific value of $\theta$. We stress that this is not equivalent to a conventional isolator, as the energy from right to left is not absorbed in the device, but rather it is reflected. Interestingly, as shown in Figure 3c, we can obtain an arbitrarily large forward transmission ~100% and complete reflection ~100% for a given range of incident power. We define the ratio between the power levels at the edges of the isolation region as the nonreciprocal intensity range (*NRIR*). Time-reversal symmetry implies a fundamental limit to NRIR for devices based on a single nonlinear Fano resonator with asymmetric decay rates to the two ports [27].

$$T \leq \frac{4 \cdot NRIx}{(1 + NRIR)^2},$$ (5)

where $T$ is the forward transmission. This is due to the fact that time-reversal symmetry requires the transmission to be identical, even in largely asymmetric resonators, if they fully transmit the input signal from one port [44]. This trade-off is plotted in Figure 1b, where the shaded blue region denotes the range of transmission and nonreciprocal intensity ranges available for any nonlinearity-based nonreciprocal device based on a single resonance satisfying the inequality (4). Interestingly, in our case, similar to previously demonstrated nonlinearity-based nonreciprocal devices based on coupled resonators [34], we do not need to comply with this limitation, and we can have an arbitrarily large forward transmission over a wide range of incident power levels. In particular, we show in our numerical results that some of the proposed designs (red asterisks in Figure 1b) can overcome the bound in Equation (5) (blue dashed region). Our approach is consistent with the method used in [34], with the difference that, in our case, we use two coupled Fano resonators, instead of a coupled Fano and Lorentzian resonator. This approach makes it easier to design the two resonators so that they have the same geometric topology with only slight deviations in the geometrical parameters for the different resonators, a property that is important for optical components where fabrication errors can easily occur and drift the parameters of the two resonators in a correlated manner. In addition, we found that it is difficult to design Fano and Lorentzian resonators similar to the approach in [34] with similar quality factors based on metasurfaces. For instance, in one of the designs we will present, the quality factor of the Fano resonance based on guided mode resonance is in the order of ~$10^3$, attained using only a thin Si layer of a few nanometers thickness~$\lambda/6$; if we try to design a Lorentzian resonant metasurface with the same quality factor based on, e.g., a Fabry–Pérot resonance, we need a homogeneous slab of Si with thickness $\sim \frac{10^3}{n} \times \lambda$, where $\lambda$ is the operation wavelength, meaning that the device will be extremely thick. Finally, we notice that the power factor $P$ scales with the nonlinear coefficient, $P \propto 1/\left|\chi^{(3)}\right|^2$; therefore, in order to obtain both a lower operating power and larger bandwidth, we can use materials with larger $\chi^{(3)}$, for example multiple quantum wells, which have been shown to support extremely large nonlinear coefficients and low incident power requirements to trigger nonlinear responses [45]. In addition, using high quality-factor Lorentzian [46] or Fano [47] resonant metasurfaces, it is possible to further boost the nonlinear response.

Based on these design requirements, we present two implementations of nonlinearity-based nonreciprocal optical metasurface devices. The first design is based on low quality-

factor metasurfaces, which are therefore less prone to fabrication errors, but requiring large intensities in the range of $GW/cm^2$. The second design is based on high quality-factor resonators, which are more sensitive to fabrication errors but require much lower input power levels. The results for the proposed designs are plotted with red asterisks in Figure 1b. Some of them lie in the red shaded region, thereby overcoming the fundamental limitation Equation (5).

### 3. Practical Implementations (for the Numerical Analysis, Please See Section 'Full Wave Numerical Simulation')

*3.1. Low Quality Factor Resonators*

As illustrated in the previous section, we designed two optical Fano resonators with shifted resonance frequencies such that they have the same unitary transmission at the same incident power level; in addition, one exhibits a bistable transmission curve, while the other smoothly varies as we increase the input power for the individual resonators. Next, we will arrange the resonators in order to realize large optical nonreciprocity.

In order to design a Fano resonator, we consider a thin dielectric slab of Si ($\epsilon_{Si} = 12$) of thickness $d_1$ with an etched groove on the top surface of thickness $d_2$ patterned periodically with period $p$, as shown in the inset of Figure 4a. We choose the period $p = \lambda/2$ to couple the normally incident wave to the guide mode of the unpatroned slab. This structure supports Fano resonances based on two coupled modes. In order to inspect the two modes in this periodic array, let us consider first the special case $d_2 = 0$, so that the thick dielectric slab acts like a Fabry–Pérot resonator with a broadband resonance, i.e., a bright mode. When we next structure the slab with periodic grooves, a normally incident plane wave, with a magnetic field polarized along the $z$ axis (TM), can couple to the dielectric waveguide mode, which is essentially a dark mode and cannot be accessible without the presence of the grooves. Upon the TM excitation, the two modes interfere, and the transmission spectrum experiences a Fano line shape as a function of the incident frequency f, as shown in Figure 4a for different parameters of $d_1$ and $d_2$. In order to confirm the role of interference between the two modes, we fit the transmission spectrum with a standard CMT model, which indeed shows excellent agreement with the full wave simulation (COMSOL Multiphysics, Stockholm, Sweden). In order to further confirm that the waveguide mode is excited, we plot the magnetic field component $H_z$, as shown in the inset in Figure 4a, which indeed shows a field distribution very similar to the $TM_{01}$ dielectric waveguide mode [42]. The quality factor of the resonance is in the order of 10, suggesting that the design is not very prone to fabrication errors and imperfections, and has a wide bandwidth.

We choose ($d_1$ and $d_2$) for each resonator, such that they satisfy the requirements illustrated in the previous section. In order to confirm this, we numerically investigate the nonlinear response of the individual resonators by including in the simulations the Kerr nonlinearity of Si, $\epsilon_{Si}(\mathbf{r}) = 12 + \chi^{(3)}|E(\mathbf{r})|^2$, where $|E|$ is the magnitude of the electric field, and $\mathbf{r}$ is the position vector. Figure 4b shows the response at the normalized resonance frequency, f/(0.5 c/p) = 1 (vertical dashed line in Figure 4a), with a high-power incident plane wave with intensity $P_{inc}$. The transmissions for two different structures with parameters shown in the legend as a function of the input intensity are plotted in Figure 4b, showing that they both exhibit very high transmission $T = -0.03$dB at the same power level, $P_{inc} = 15\,GW/cm^2$, as shown in the inset at point ℵ, confirming the design requirements presented in the previous section.

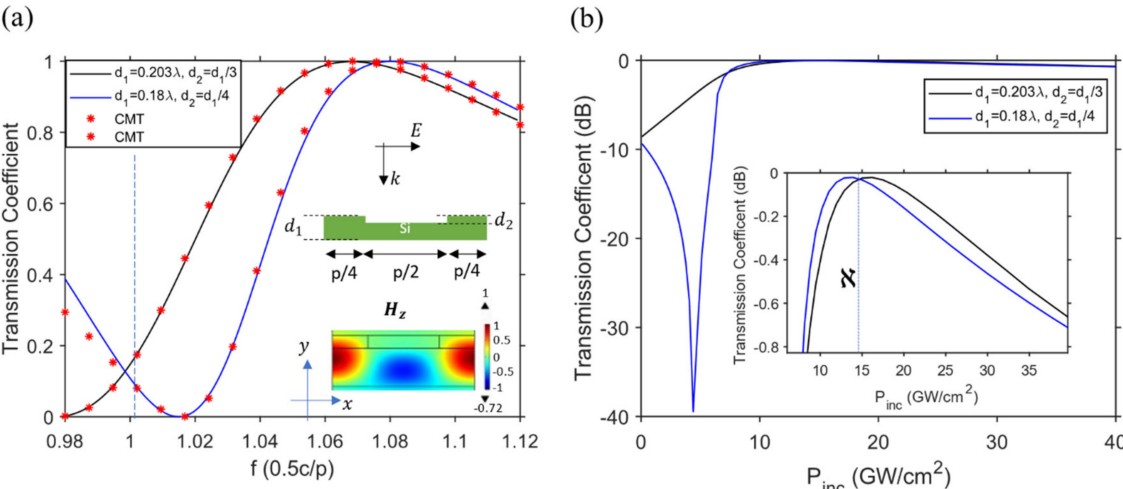

**Figure 4.** Full wave simulation results for the design of an optical Fano resonator for free-space radiation at normal incidence, achieved by choosing $p = \lambda/2$, where $\lambda$ is the free-space wavelength. (**a**) Linear response of the transmission coefficient of individual resonators arranged in a periodic array, with period $p$ excited with a normally incident plane wave, with polarization as shown in the inset. The red markers indicate the CMT results with the fitting parameters given in Table 3 in Section 5.5 (Fano Resonator Parameters). The color plot inset shows the z-component of the magnetic field distribution inside the slab, consistent with the guided mode inside the unperturbed dielectric slab of the same thickness, and the color bar is normalized to the maximum field. (**b**) Transmission coefficient for individual resonators in dB versus the incident power for normal incidence assuming $\chi^{(3)} = 2.8 \times 10^{-18} m^2/V^2$ at the resonance frequency.

We can now form a nonreciprocal device by arranging the two periodic resonators back-to-back, separated by an electrical length $\theta$ (a top grooved slab with $d_1 = 0.18\lambda_0$ and $d_2 = d_1/4$, separated by a thickness $l = \lambda\theta/(2\pi)$ and a bottom grooved slab with $d_1 = 0.203\lambda_0$ and $d_2 = d_1/3$, $p = \lambda_0/2$ where $\lambda_0 = 1.55$ μm, and $\theta = 1.5\pi$), where one unit cell is shown in the inset of Figure 5a. We consider high power excitation ($H_z$) with increased input intensities ($P_{inc}$) from top and bottom, and plot the transmission coefficients $T_{12}$ and $T_{21}$ at different operating wavelengths. Figure 5a shows the transmission coefficient when excited from different sides at the operation wavelength $\lambda_0$. As expected, at a low incident power $P_{inc} < 10.5$ GW/cm$^2$, the system is almost reciprocal, resulting in $T_{12} \approx T_{21}$. As we increase the incident power, specifically at $P_{inc} = 10.5$ GW/cm$^2$, $T_{21}$ experiences a sharp transition from 0 to 0.95, while at the same time $T_{21}$ remains very low. As we further increase the input intensity, the transmission $T_{12}$ exhibits a similar abrupt transition from 0.07 to 0.98. The intensity range between the two transitions defines our NRIR = 0.27 dB, and the highest transmission between the two transitions is $-0.132$ dB, plotted in Figure 1b, showing that at wavelength $\lambda_0 = 1.55$ μm the device does not overcome the bounds of an optimal single resonance device.

We can also calculate the transmission at larger wavelengths $\lambda_0 = 1.56$ μm and at $\lambda_0 = 1.565$ $\mu m$, as shown in Figure 5b,c, respectively. Interestingly, the response at $\lambda_0 = 1.565$ $\mu m$ shows an NRIR = 0.984 dB and the highest transmission $T = -0.037$ $dB$, and it indeed shows that the response breaks the limitation of a single Fano nonlinear device, as denoted by its asterisk point in Figure 1b. However, the design at $\lambda_0 = 1.56$ $\mu m$ still satisfies the bounds for a single nonlinear resonator, as indicated in Figure 1b. We also calculated the transmission in the dB scale, finding that $T_{12} = -50$ dB at the peak transmission for $T_{21}$ when $\lambda = 1.565$ μm, demonstrating a large contrast in transmission, a sort of pseudo-isolation of 50 dB for this device at an incident power $P_{inc} = 18$ GW/cm$^2$. We stress that we should not consider this quantity as a conventional isolation metric, because it is based on the assumption that the two ports are excited independently, and the device cannot fully isolate in the general case of continuous wave excitation from opposite ports. At this level of incident power and frequency, we show the full-wave simulation results of the field distribution when the device is excited from each side in the inset of

Figure 5c, which indeed shows close to unitary transmission in the forward propagation, while the field is largely reflected in backward propagation, forming a standing wave at the opposite port. Notice that this functionality is very different from a conventional isolator, which would instead necessarily absorb the input energy coming from the input port. This reflectivity can be used to our advantage, for instance by routing it to a third port for circulation purposes [1]. Overall, the results in Figure 5 are far superior to those reported in [23,24] for designs using the same material parameters; however, here we use larger input intensities because of the low-quality factor of the involved resonators. As shown in Figure 5a–c for each frequency, the device works for slightly different power levels. Therefore, in order to characterize the response in the frequency domain at a fixed input power, we plot in Figure 5d the transmission versus the frequency at a fixed input power of 16.5 GW/cm$^2$, which shows the isolation of the bandwidth greater than 600 GHz around the telecommunication wavelength, confirming the wideband response of the device due to the low quality factor of the constituent resonators, at the cost of increased required power levels.

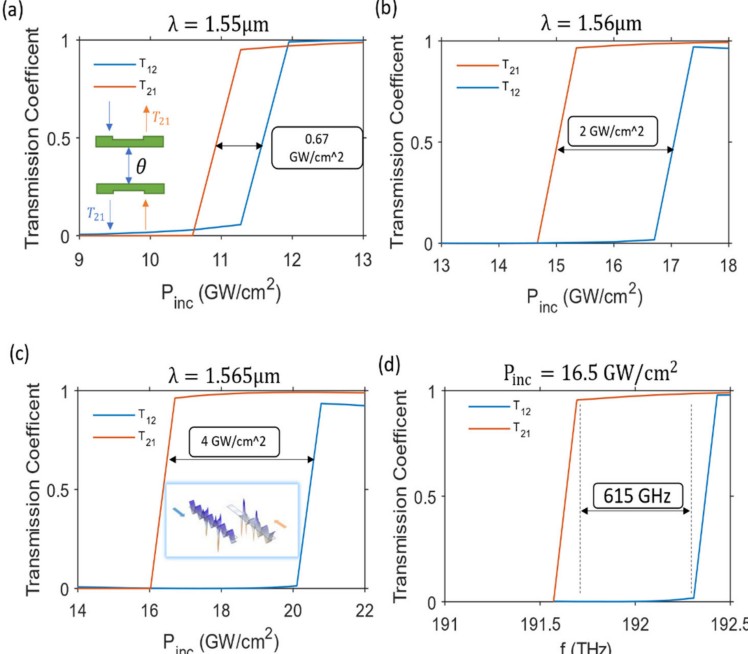

**Figure 5.** Full wave simulation results for transmission from opposite ports ($T_{12}$, $T_{21}$) of the proposed bilayer metasurface shown in the inset of (**a**), where the top resonator has dimensions ($d_1 = 0.203\lambda$, $d_2 = d_1/4$) and the bottom resonator has dimensions ($d_1 = 0.18\lambda$, $d_2 = d_1/3$) for the electrical distance $\theta = 1.5\pi = 2\pi/\lambda l$, and l is the physical distance between the two resonators. The period of the two-groove array is $\lambda/2$ where $\lambda = 1.55$ μm. The transmission of a normally incident plane wave with increased input power $P_{inc}$ at (**a**) wavelength $\lambda_0 = 1.55$ μm, (**b**) $\lambda_0 = 1.56$ μm and (**c**) $\lambda_0 = 1.565$ μm. The inset in (**c**) shows the full wave simulation of the magnetic field when the device is excited from each side separately when $P_{inc} = 18$ GW/cm$^2$. (**d**) Transmission from two sides versus the frequency at a fixed power level of 16.5 GW/cm$^2$. The NRIR and transmission for the plots from (**a**) to (**c**) are NRIR = [0.27, 0.541, 0.984] dB, and the peak transmission is T = [−0.132, −0.04, −0.037] dB, respectively.

### 3.2. High Quality Factor Resonators

As shown in the previous section, the power levels required for the device to work in its optimal condition are in the order of GW/cm$^2$, due to the low-quality factor of the resonator, which may hinder its broad applicability. However, one possible solution is to employ materials with high nonlinearity, e.g., multiple quantum wells have shown extremely high nonlinear susceptibilities both in both the second and third orders [45,48,49].

Higher-Q Fano resonators can be used to reduce the required power, however, and for this reason we aim to maximize the field enhancement levels inside the resonators by designing a high-Q Fano resonator based on guided mode resonance coupling through narrow slits in a dielectric slab, as shown in inset of Figure 6a. The transmission coefficient for a normally incident plane wave, with an electric field in the z-direction (TE), is shown in Figure 6a, exhibiting a typical Fano line shape with zero transmission at $\lambda_0 = 1.5608$ μm and unitary transmission at the very nearby wavelength $\lambda_0 = 1.5598$ μm. In order to evaluate the quality factor of this resonator, we plot the average electric field enhancement inside the nonlinear material $|E|^2$, as shown in Figure 6b, which shows a Lorentzian line shape with quality factor $Q = 1700$, compared to $Q = 10$ of the design in the previous section. Additionally, the inset shows that the electric field distribution inside the slab is similar to the $TE_{01}$ waveguide mode [42]. The second resonator is formed by adding a glass slab ($n_{glass} = 1.5$) to another Si grating, as shown in the inset of Figure 6c. Similar to Figure 6a, the transmission exhibits a sharp Fano resonance; however, the average electric field enhancement is smaller than in Figure 5d. This is understood, as we chose the guided mode resonance to be formed inside the glass slab, and not in the Si layer, as confirmed by the electric field distribution in the inset of Figure 6d. We chose the second resonator to be attached to the glass substrate in order to control the field enhancement in the Si layer and hence obtain unitary transmission at the same power levels when the nonlinearity is included (similar to point ℵ in Figure 4b). In addition, we plot the CMT standard model results in Figure 6a, and Figure 6c using the fitting parameters from Table 3 in Section 5.5 (Fano Resonator Parameters), which shows excellent agreement with the full wave simulation.

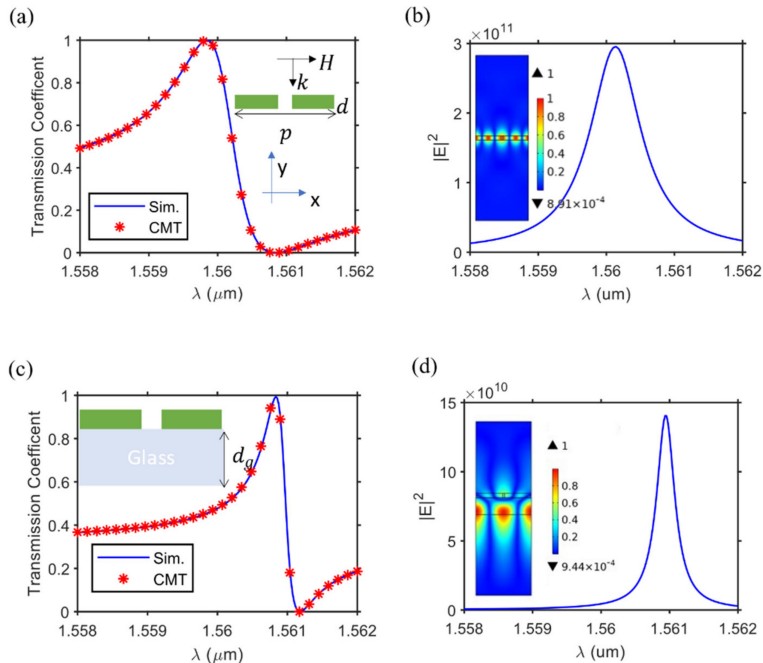

**Figure 6.** Full wave simulation results for the linear response of the Fano resonators. (**a**) Linear transmission of a periodically patterned dielectric slab for a normally incident plane wave excitation with an electric field along the z axis. The red markers are the CMT standard model results using the parameters given in Table 3 (see Section 5.5 (Fano Resonator Parameters)) for resonator 1 (*w/o* glass). (**b**) Average squared value of the electric field inside the Si material shown in (**a**); the inset shows the electric field distribution at resonance, and the scale bar is normalized to the maximum field. (**c**) The same as (**a**), but with an added glass slab. The red markers are the CMT standard model results using the parameters given in Table 3 (see Section 5.5 (Fano Resonator Parameters)) for resonator 2 (w glass). (**d**) The same as (**b**) but for the structure in (**c**). The geometry parameters used are d = 0.1179 μm, *p* = 1.4763 μm, dg = 0.45875 μm, and slit width = 0.0536 μm.

By exciting each metasurface separately with a normally incident TE plane wave with input intensity $P_{inc}$, we can calculate the transmission as a function of the intensity, considering the Si nonlinearity using full wave simulation (COMSOL Multiphysics). The nonlinear transmission coefficient is shown in Figure 7a for the two resonators independently. We stress that the excitation for each structure from different sides results in a symmetric transmission curve following identically the same curves in Figure 7a, essentially having NRIR = 0 dB. Following a similar design process as above, we are able to achieve unitary transmission for each individual resonator at the same incident power level $P_{inc} = 1.5$ MW/cm$^2$ (point ℵ in the inset of Figure 7a), which is much smaller than the input intensity of 15 GW/cm$^2$ obtained in the previous section. Additionally, we can plot the CMT results for this nonlinear behavior, which shows excellent agreement with the numerical simulation. We can form an optimal nonreciprocal device by stacking the two gratings of Figure 7a, separated by the distance $l = \lambda\theta/(2\pi)$, where $\lambda = 1.56104$ µm and $\theta = 6.4\pi$, as shown in the inset of Figure 7b. The nonlinear transmission coefficient for excitation from opposite sides for increased input intensities $P_{inc}$ is shown in Figure 7b. We obtain very high transmission $T = -0.2$ dB and large transmission contrast of over 60 dB at an incident power level of only 1.4 MW/cm$^2$. Interestingly, the NRIR = 2.1 dB, and by plotting the point $(NRIR, T)$ in Figure 1b, we see that this design exceeds the limitation of a single Fano nonlinear resonator. We emphasize that this power level is very small, and can be achieved even with continuous wave lasers; however, this comes at the cost of being vulnerable to fabrication errors [50]. Concepts borrowed from topological photonics may be used to enhance the robustness of the response in high-quality factor systems, even in the presence of fabrication errors [51,52].

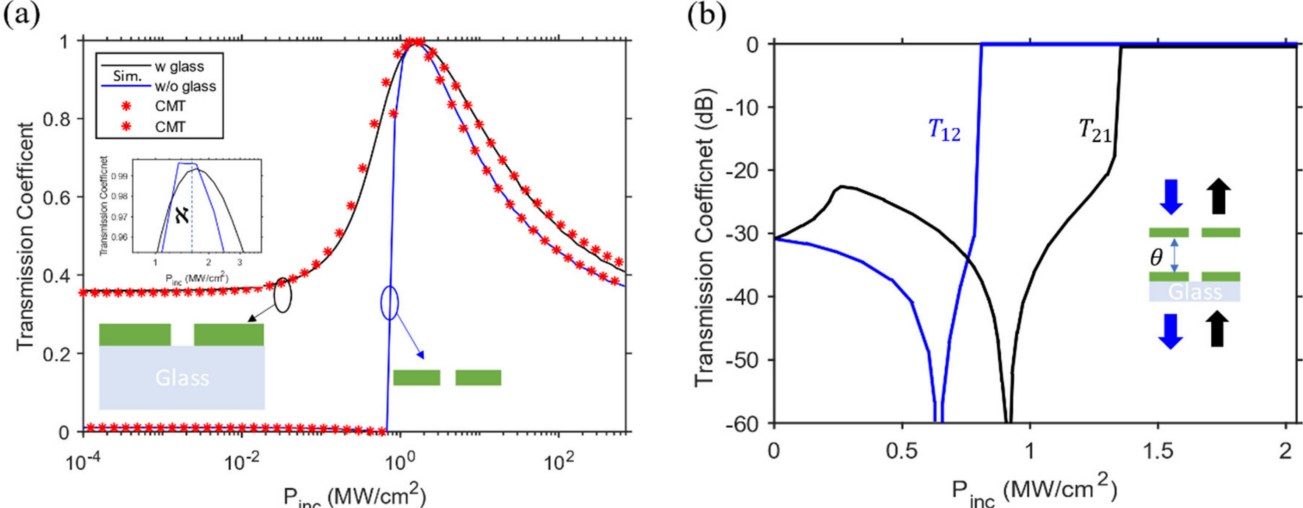

**Figure 7.** Full wave simulation results for excitation at frequency c/1.56104 [um], with a normally incident plane wave of power $P_{inc}$. (**a**) Nonlinear transmission for isolated gratings as a function of the input power. The red markers indicate the results obtained using the CMT standard model with the parameters given in Table 3 (see Section 5.5 (Fano Resonator Parameters)). (**b**) Transmission for the coupled metasurface design with $\theta = 6.4\pi$ shown in the inset. $T_{12}$ indicates the transmission from top to bottom, while $T_{12}$ indicates the transmission from bottom to top. The NRIR is 2.1 dB, and the peak forward transmission is $-0.2$ dB.

We also studied the response of this optimal device for different electrical lengths $\theta$ in Figure 8. Due to the nonlinearity of the system, the response is interestingly not necessarily periodic, even after the evanescent field effects disappear. At very large input powers, past the bistable response region, the response becomes periodic, as shown for the case $\theta = 10.4\pi$, which is very similar to $\theta = 6.4\pi$ in Figure 7b. Additionally, as illustrated above, the value of $\theta$ largely controls the direction of isolation; for example, the values of $\theta$ ($\theta = 3.6, 4.4, 5.2, 10.4\pi$) give forward transmission and backward isolation, and the

converse is true for the other values of $\theta = 3.8\pi, 4.8\pi$. Additionally, there are some values of $\theta$ where the response becomes symmetric and the field enhancement becomes symmetric for different directions of propagation ($\theta = 4\pi, 4.2\pi$).

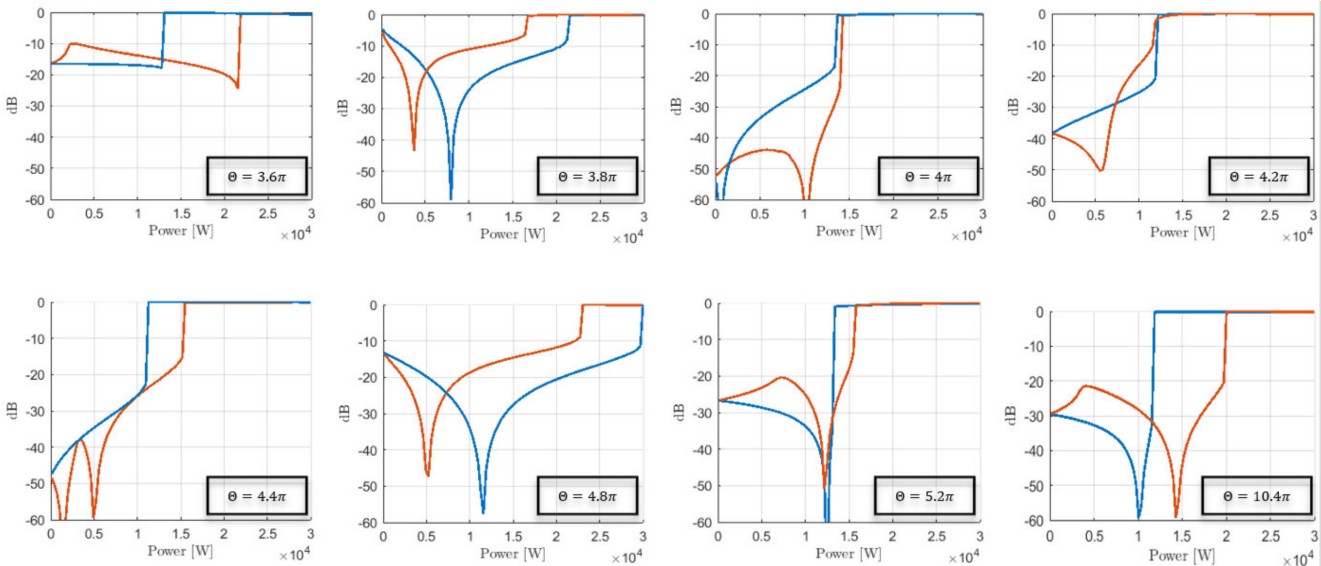

**Figure 8.** Full wave simulation results for the nonlinear device in the inset of Figure 7b, excited at frequency c/1.56104 [um] with a normally incident plane wave of power $P_{inc}$ for the different values of θ displayed in the bottom right corner of each figure. The horizontal axis is normalized by the period $p = 1.475$ μm such that it has units of Watts.

We can characterize the operation of the proposed metasurface by comparing it to other free-space nonreciprocal metasurface devices operated under different principles. In Table 2, we present a comparison highlighting the main aspects characterizing the response of these different nonreciprocal metasurfaces, focusing on the main approaches used to break the reciprocity, i.e., breaking time invariance [11], passivity [17] and linearity [23], including this work.

As seen in Table 1, we notice that the bandwidth of all of the proposed approaches is very small, and it does not exceed a 5% fractional bandwidth in the best case [19]. This can be illustrated because all of the approaches employ resonant elements with a high quality factor in order to reduce the power requirements, and as a consequence decrease the bandwidth. We also observe that it is often difficult to characterize the efficiency of a metasurface at RF frequencies because there is a non-trivial number of parameters that need to be taken into account in order to estimate the total power efficiency. For example, in time-modulated metasurfaces, the efficiency should be calculated by normalizing the output power to the total input power, including the modulation power. In fact, it is very valuable that the authors in [13] did a full characterization of the metasurface performance by calculating all of the relevant power quantities. Although the insertion loss in the their work is about $-5$dB, this does not mean that the efficiency is low compared to other works based on the same concept, because other works [11,14] do not provide a full estimation of the relevant power quantities. However, we emphasize that for nonlinear metasurfaces, it is very straightforward to characterize the overall efficiency because the signal self-modulates the medium through the Kerr nonlinearity, and the metasurface itself is passive.

**Table 1.** Comparison between the different approaches used to realize free space nonreciprocity.

| Work | Breaking Reciprocity Due to: | Bandwidth/ Center Frequency | Modulation Frequency (mf) or Total Gain of the Amplifiers (tga) or Kerr Nonlinearity Coefficient ($\chi^{(3)}$) | | Thickness/ Wavelength | Pump Power Per Unit Cell or Signal Power/Intensity | Isolation (or Transmission Contrast) at Best Insertion Loss | Frequency Conversion/ Programmable |
|---|---|---|---|---|---|---|---|---|
| [11] | Time modulation | NA/ 5.28 [GHz] | MF | 50 [MHz] | 2.54 [mm]/ 56.8 [mm] | Modulation signal power or intensity | NA | No/yes |
| [13] | | 0.3 [GHz]/ 8.97 [GHz] | | 370 [MHz] or 600 [MHz] | 2 [mm]/ 33.33 [mm] | 10 dBm or 1 V | 5 dB loss, isolation of 30 dB | Yes/yes |
| [14] | | 5.77 [THz]/ 348.8 [THz] | | 2.8 [THz] | 400 [nm]/ 860 [nm] | 15 GW/cm² | NA | Yes/yes |
| [17] | Unidirectional gain amplifiers | 6 [MHz]/ 944 [MHz] ¥ | TGA | 0 dB | 31.7 [mm]/ 317 [mm] | DC power of each amplifier in one layer, number of layers | NA, 2 | Isolation of −1.5 dB assuming 0 dB insertion loss ¥ | No/yes |
| [18] | | 0.17 [GHz]/ 5.9 [GHz] | | 20 dB | 1.7 [mm]/ 50.8 [mm] | 0.18 [W], 2 | 17 dB transmission gain and 10 dB loss correspond to 27 dB. | |
| [19] | | 0.25 [GHz]/ 5.875 [GHz] | | 10 dB-30 dB | 1.82 [mm]/ 51 [mm] | 0.1–0.2 [W] £, 2 | 20 dB transmission gain and 20 dB loss correspond to 40 dB isolation | |
| [20] | | 0.05 [GHz]/ 5.5 [GHz] | | 20 dB | 2.1 [mm]/ 54.54 [mm] | 0.1 [W], 2 | 13 dB of transmission gain and 32 dB isolation | |
| [23] | Kerr nonlinearity | NA | $\chi^{(3)}$ (m²/V²) | 2.8 × 10⁻¹⁸ | 0.1 [um]/1.5 [um] | Signal intensity | 5 kW/cm² | −17 dB at −1.2 dB over 4.77 dB * | No/no |
| [24] | | | | | (2.7–6.15) [um]/ 1.53 [um] | (1.5–2) MW/cm² | Isolation of −25.4 dB at insertion loss of −0.46 dB over NRIR of 2.79 dB ** Isolation of −35.7 dB at insertion loss of −0.41 dB over NRIR of 1.5 dB *** Isolation of −15.2 dB at −0.044 dB over NRIR of 1.52 dB ****¿ | |
| This work | | 0.6 [THz]/ 192 [THz] | | | (1.33–5.334) [um]/ 1.56 [um] | (16.8–0.001) GW/cm² | −56 dB at −0.04 dB, −65 dB at −0.2 dB | |

¥ From Figure 4 [16]. £ Based on the datasheet of the amplifier and the DC bias used in the paper and similar transistors used in previous publications [17], it is expected to be 0.1–0.2 W [18]. * From Figure 3c [23]. ** From Figure 3 [22]. *** From Figure 4 [22]. **** From Figure 5 [22]. ¿ Notice that the NRIR of this metasurface does not break the limitation of the single Fano resonator nonlinear isolator, even though 1.52 dB NRIR goes beyond the limit, the isolation is not infinite, so a fair comparison cannot be guaranteed here.

## 4. Conclusions

We studied and designed nonlinearity-based nonreciprocal metasurface devices based on coupled Fano resonances. Their response has been shown to overcome the fundamental limitations of a single resonant element associated with time-reversal symmetry [27,44], and our designs show an insertion loss of $-0.04$ dB for an NRIR of 1 dB with isolation over 50 dB. Our results are presented in normalized units, such that they can be scaled to other frequency ranges, and even applied to other materials, such as multiple quantum wells supporting larger nonlinearities [45]. We focused on two designs, with one supporting a moderate resonance and therefore requiring large power levels, but with robust response to fabrication errors. The second design is more prone to fabrication errors, but it requires much smaller power levels to operate. It is, of course, possible to explore designs that work in between these two extremes, as a function of the desired trade-off between power levels and the robustness of the design. It is interesting that all of the considered designs, while surpassing the bound for single resonant elements in Equation (5), lie very close to it, as seen in Figure 1b. It appears that metasurface designs do not provide the same flexibility of circuit elements, which provided a superior performance in terms of NRIR [34]. Our work provides useful guidelines to design these types of nonlinearity-based nonreciprocal metasurfaces for free-space radiation. Similar concepts can also be extended to photonic integrated circuits for which the control over the structure parameters is easier, and it provides a useful platform for passive, bias-free, magnetic-free CMOS-compatible nonreciprocal components for optical communications, sensing, imaging, and computing.

## 5. Materials and Methods

### 5.1. Coupled Mode Theory

In order to illustrate the formation of a Fano resonance, we consider in Figure 9 two generic resonant modes coupled to radiation channels. We assume a complex field inside resonator $n$, $a_n$ such that $|a_n|^2$ is the normalized energy inside the resonator. We derive the CMT equations as

$$\frac{da_1}{dt} = (j\omega_1 - \kappa_1)a_1 + \sqrt{\kappa_1}(S_{2+} + S_{1+})$$
$$\frac{da_2}{dt} = (j\omega_2 - \kappa_2)a_2 + \sqrt{\kappa_2}(S_{2-}e^{-j\theta} + S_{3+})$$
$$S_{1-} = \sqrt{\kappa_1}a_1 - S_{2+}$$
$$S_{2-} = \sqrt{\kappa_1}a_1 - S_{1+}$$
$$S_{2+} = e^{-j\theta}(\sqrt{\kappa_2}a_2 - S_{3+})$$
$$S_{3-} = (\sqrt{\kappa_2}a_2 - S_{2-}e^{-j\theta}).$$

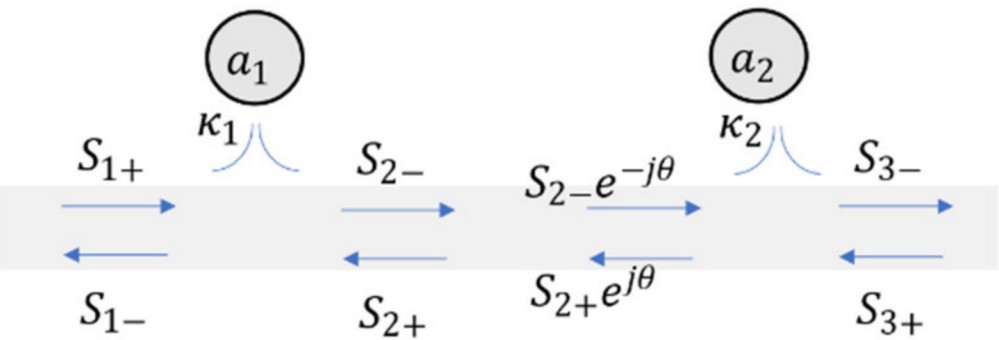

**Figure 9.** Schematic model of coupled resonators to and from a Fano resonance.

Assuming excitation from the left port with the form $S_{1+} = e^{j\omega t}$, we rewrite the steady-state equations, assuming that all modes and signals have the form $A \to A e^{j\omega t}$, as

$$(j\delta_1 + \kappa_1)a_1 - \sqrt{\kappa_1}S_{2+} = \sqrt{\kappa_1}$$
$$(j\delta_2 + \kappa_2)a_2 - \sqrt{\kappa_2}S_{2-}e^{-j\theta} = 0$$
$$S_{1-} - \sqrt{\kappa_1}a_1 + S_{2+} = 0$$
$$S_{2-} - \sqrt{\kappa_1}a_1 = -1$$
$$S_{2+} - e^{-j\theta}\sqrt{\kappa_2}a_2 = 0$$
$$S_{3-} - \sqrt{\kappa_2}a_2 + S_{2-}e^{-j\theta} = 0$$

where, $\delta_n = \omega - \omega_n$. Then, we solve for $a_1$ and $a_2$, and then substitute in $S_{3-}$ to obtain the transmission coefficient.

$$a_1 = \frac{j\delta_2\sqrt{\kappa_1}}{j(\delta_2\kappa_1 + \delta_1\kappa_2) - \delta_1\delta_2}S_{1+}; \; |a_1|^2 = \frac{\frac{\kappa_1^2}{\delta_1^2}\frac{1}{\kappa_1}}{1 + \left(\frac{\kappa_1}{\delta_1} + \frac{\kappa_2}{\delta_2}\right)^2}|S_{1+}|^2. \tag{6}$$

$$a_2 = \frac{-j\delta_1\sqrt{\kappa_2}}{j(\delta_2\kappa_1 + \delta_1\kappa_2) - \delta_1\delta_2}S_{1+}; \; |a_2|^2 = \frac{\frac{\kappa_2^2}{\delta_2^2}\frac{1}{\kappa_2}}{1 + \left(\frac{\kappa_1}{\delta_1} + \frac{\kappa_2}{\delta_2}\right)^2}|S_{1+}|^2. \tag{7}$$

$$t = \frac{S_{3-}}{S_{1+}} = -\frac{1}{1 - j\left(\frac{\kappa_2}{\delta_2} + \frac{\kappa_1}{\delta_1}\right)}; \; T = |t|^2 = \frac{1}{\left(\frac{\kappa_1}{\delta_1} + \frac{\kappa_2}{\delta_2}\right)^2 + 1} = \frac{1}{1 + x_a^2}. \tag{8}$$

$$r = \frac{S_{1-}}{S_{1+}} = 1 - t = \frac{-j\left(\frac{\kappa_2}{\delta_2} + \frac{\kappa_1}{\delta_1}\right)}{1 - j\left(\frac{\kappa_2}{\delta_2} + \frac{\kappa_1}{\delta_1}\right)}; \; R = |r|^2 = \frac{\left(\frac{\kappa_1}{\delta_1} + \frac{\kappa_2}{\delta_2}\right)^2}{\left(\frac{\kappa_1}{\delta_1} + \frac{\kappa_2}{\delta_2}\right)^2 + 1} = \frac{x_a^2}{1 + x_a^2}. \tag{9}$$

There are two zeros in transmission, when $\delta_1 = 0$ or $\delta_2 = 0$. Furhtermore, the unitary transmission happens at $\omega = \frac{\kappa_2\omega_1 + \kappa_1\omega_2}{\kappa_1 + \kappa_2}$, or equivalently when $\frac{\kappa_1}{\delta_1} = -\frac{\kappa_2}{\delta_2}$. Throughout the work, we assume that the second resonance frequency is much larger than the first resonance frequency $\omega_2 \gg \omega_1$, with a much higher coupling rate, $\kappa_2 \gg \kappa_1$.

*5.2. Nonlinear Bistability*

If we assume the resonators to be nonlinear, and $\omega_2$ is far away from the frequency range of interest, the resonance frequency of the mode $a_1$ changes according to the relation [42],

$$\omega_1 = \omega_0\left(1 - \frac{|a_1|^2}{|a_0|^2}\right), \tag{10}$$

where $\omega_0$ is the resonance frequency of the mode $a_1$ before including nonlinearity, and $|a_0|^2$ quantifies the nonlinearity, specifically, the rate of the resonance frequency shift due to nonlinearity. We can substitute for $|a_1|^2$ from (10) into (6) and obtain

$$\delta_1 = \delta_{01} + \frac{\left(\frac{\kappa_1}{\delta_1}\right)^2}{1 + \left(\frac{\kappa_1}{\delta_1} + \frac{\kappa_2}{\delta_2}\right)^2}\frac{\omega_{01}|S_{1+}|^2}{\kappa_1|a_0|^2}, \tag{11}$$

where, again, $\delta_{01} = \omega - \omega_0$. This is a cubic equation of the only unknown $\delta_1$ when all other parameters are known, and it has the form

$$\left(\frac{\kappa_1}{\delta_1}\right)^3(P + \updownarrow) + \left(\frac{\kappa_1}{\delta_1}\right)^2(2\updownarrow\dagger - 1) + \left(\frac{\kappa_1}{\delta_1}\right)\left(\updownarrow + \updownarrow\dagger^2 - 2\dagger\right) - 1 - \dagger^2 = 0 \tag{12}$$

where the shorthand parameters $(P, \Updownarrow, \dagger)$ are given by

$$P = \frac{\omega_0 |S_{1+}|^2}{\kappa_1 |a_0|^2}, \Updownarrow = \frac{\delta_{01}}{\kappa_1}, \dagger = \frac{\kappa_2}{\delta_2}$$

By solving (12), we obtain the shift in resonance frequency including nonlinearity, which can later be substituted in (8) to obtain the nonlinear transmission coefficient.

### 5.3. Bistability Condition

The bistability of a single nonlinear Lorentz resonator can be achieved at a certain range of power levels if the resonator is excited with a frequency $\omega$, such that [42]

$$\omega < \omega_{0L} - \sqrt{3}\,\kappa_{1L}, \tag{13}$$

where $\omega_{0L}$ is the resonance frequency of the resonator at which the maximum energy is stored when exited; $\kappa_{1L}$ is its total decay rate, and we added the subscript $L$ to refer to Lorentz resonator. We can derive a similar condition for our two-mode Fano resonator. This can be accomplished by writing the energy in $a_1$ of our Fano resonator in a similar form to the energy of a Lorentz resonator, and then finding the resonance frequency $\omega_{\mathrm{mod}}$ and coupling rate $\kappa_{\mathrm{mod}}$ that are equivalent to $\omega_{0L}$ and $\kappa_{1L}$. In order to derive these effective parameters, we start from the fact that the Fano resonant modes satisfy the relation $|\omega_2 - \omega_1| \gg \delta_1$, which means that we are interested in a solution in the vicinity of the high-quality mode $a_1$. Therefore, a good approximation is

$$\delta_2 = \omega - \omega_2 \approx \omega_1 - \omega_2 = \Delta_\omega.$$

Then, we rewrite the mode energy $|a_1|^2$ from Equation (6) as

$$|a_1|^2 \approx \frac{\frac{\kappa_1 \Delta_\omega^2}{(\Delta_\omega^2 + \kappa_2^2)}}{\left(\delta_1 + \frac{\Delta_\omega \kappa_1 \kappa_2}{(\Delta_\omega^2 + \kappa_2^2)}\right)^2 + \frac{\Delta_\omega^4 \kappa_1^2}{(\Delta_\omega^2 + \kappa_2^2)^2}} |S_{1+}|^2 \tag{14}$$

It is easily shown from Equation (14) that the energy inside the Fano resonator follows a Lorentzian lineshape, such that we can proceed to derive the equivalent quantities in order to derive the bistability condition. Additionally, we can see that the largest energy is achieved at a frequency that lies between the zero and unitary transmission of the Fano resonator. Thus, this frequency is equivalent to $\omega_{0L}$, and it is given by

$$\omega_{\mathrm{mod}} = \omega_1 - \frac{\Delta_\omega \kappa_1 \kappa_2}{\Delta_\omega^2 + \kappa_2^2}. \tag{15}$$

Recall that $\Delta_\omega < 0$ so $\omega_{\mathrm{mod}} > \omega_1$. We can also derive the effective coupling rate (linewidth) for this lineshape as

$$\kappa_{\mathrm{mod}} = \frac{\Delta_\omega^2 \kappa_1}{\Delta_\omega^2 + \kappa_2^2}. \tag{16}$$

Therefore, the condition for bistabitlity in the proposed Fano resonator is given by

$$\omega < \omega_{\mathrm{mod}} - \sqrt{3}\kappa_{\mathrm{mod}}. \tag{17}$$

### 5.4. Effective Power of Coupled Nonlinear Fano Resonators

When we connect two Fano resonators $(a, b)$ with shifted resonance frequencies, as shown in Figure 10, we can calculate the incident power to the right Fano resonator when the device is excited from the left with incident power $|S_{1a+}|^2$. It is worth mentioning

that each Fano resonator in Figure 10 can be modeled as a coupled mode resonator, as shown in Figure 9, and that each resonator can be characterized by its scattering matrix, for instance, resonator $(a, b)$ to the (left/right) has the field reflection coefficient $(r_a, r_b)$ and transmission coefficient $(t_a, t_b)$. We calculate $P_{\text{eff}} = |S_{1b+}|^2$ under the condition that we operate the device in power levels that ensure complete reflection from the second resonator, i.e., $S_{1b-} = e^{j\phi_b} S_{1b+}$, where $\phi_b$ is the reflection phase from the second resonator. In addition, we assume that the first resonator reflection coefficient in the form $r_a e^{j\phi_a}$, which is almost constant with increasing power levels. We can easily write the following equations:

$$S_{2a-} = t_a S_{1a+} + |r_a| e^{j\phi_a} S_{2a+}$$

$$S_{2a-} = t_a S_{1a+} + |r_a| e^{j(\phi_a)} e^{-j\theta} S_{1b-}$$

$$S_{2a-} = t_a S_{1a+} + |r_a| e^{j(\phi_a+\phi_b)} e^{-j2\theta} S_{2a-}$$

$$S_{2a-} = t_a S_{1a+} + |r_a| e^{j(\phi_a+\phi_b)} e^{-j2\theta} S_{2a-}$$

$$S_{2a-} \left( 1 - |r_a| e^{j(\phi_a+\phi_b)} e^{-j2\theta} \right) = t_a S_{1a+}$$

$$S_{2a-} = \frac{t_a}{1 - |r_a| e^{j(\phi_a+\phi_b)} e^{-j2\theta}} S_{1a+}$$

$$|S_{2a-}|^2 = |S_{1b+}|^2 = P_{\text{eff}} = \frac{T_a}{1 + R_a - 2|r_a| \cos(\phi_a+\phi_b+2\theta)} |S_{1a+}|^2$$

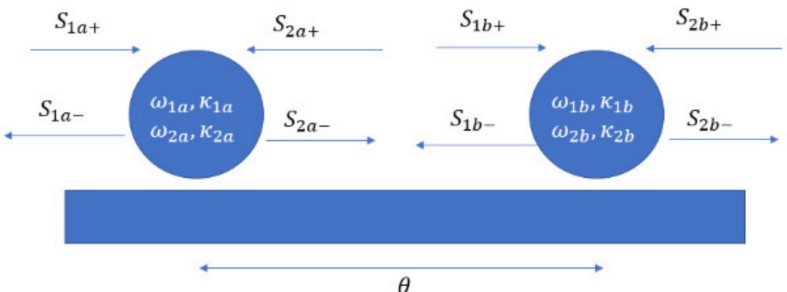

**Figure 10.** Coupled Fano resonators with shifted zero-transmission frequencies.

We use the relation $|r_a| = \sqrt{R_a} = \sqrt{1 - T_a}$, and define $|S_{1a+}|^2 = P$, so finally:

$$P_{\text{eff}} = \frac{T_a}{2 - T_a - 2\sqrt{1 - T_a} \cos(\phi_a + \phi_b + 2\theta)} P \tag{18}$$

*5.5. Fano Resonator Parameters for Figures 3, 4, 6 and 7*

We can interpret the Fano resonances from the structured Si layers used throughout this work based on their definition as the interference of a broadband bright resonance and a narrowband dark resonance. The dark resonance can be coupled to the background continuum when it is coupled evanescently to some bright mode, or through diffraction by perturbing the mode parameters, as in our case. In the case of guided mode resonance, the bright resonance is the Fabry–Pérot resonance supported longitudinally by the unperturbed dielectric slab, while the dark resonance is the guided mode of the dielectric slab, which is perturbed by the thin grooves in the dielectric. These two modes couple to the background with different coupling coefficients, and indeed they have different resonance frequencies. In order to derive the result in Figure 4, we find the best fitting parameters as given in Table 2. It is interesting to see that the value of the coupling coefficient of the dark mode $\kappa_1$ is smaller than the coupling of the bright mode $\kappa_2$. Table 3, on the other hand, gives the fitting parameters for Figures 3, 6 and 7, which also show that the coupling coefficient of the dark mode is much smaller than that of the bright mode.

**Table 2.** Parameters of the Fano resonance in Figure 4.

| Structure 1 ($d_1 = 0.18\lambda, d_2 = d_1/3$) ($\omega$, $\kappa$) are normalized by ($2\pi \times c/2p$) | $\omega_1 = 1.016$ | $\omega_2 = 1.3$ | $\kappa_1 = 0.04$ | $\kappa_2 = 0.138$ | $\theta = 0$ |
|---|---|---|---|---|---|
| Structure 2 ($d_1 = 0.203\lambda, d_2 = d_1/4$) ($\omega$, $\kappa$) are normalized by ($2\pi \times c/2p$) | $\omega_1 = 0.977$ | $\omega_2 = 1.33$ | $\kappa_1 = 0.07$ | $\kappa_2 = 0.2$ | $\theta = 0$ |

**Table 3.** Parameters of the Fano resonance in Figures 3, 6 and 7 (operating frequency = $0.9999 \times 1.92162 \times 10^{14} \times 2\pi$ (rad/s)) *.

| Structure 1 (w/o glass) ($\omega$, $\kappa$) are normalized by ($2\pi \times 10^{14}$) | $\omega_1 = 1.92204$ | $\omega_2 = 100$ | $\kappa_1 = 0.00191$ | $\kappa_2 = 152.2$ | $\theta = 0$ | $|a_0|^2 = 4.5 \times \frac{\omega_1}{\kappa_1^2}$ |
|---|---|---|---|---|---|---|
| Structure 2 (w glass) ($\omega$, $\kappa$) are normalized by ($2\pi \times 10^{14}$) | $\omega_1 = 1.921628$ | $\omega_2 = 100$ | $\kappa_1 = 0.00062$ | $\kappa_2 = 144.4$ | $\theta = 0$ | $|a_0|^2 = 10 \times \frac{\omega_1}{\kappa_1^2}$ |

* Notice that the numbers 4.5 and 10 in the last column have units of Watts.

### 5.6. Full Wave Numerical Simulation (We Used Full Wave Numerical Simulation to Obtain the Results in Figures 4–8)

The characterization of the metasurface was performed using linear and nonlinear simulations in the commercial software COMSOL Multiphysics. They are based on finite element numerical simulations, where the solution domain is divided into nonuniform free triangular elements, and Maxwell's equations are solved inside and on the boundaries of these elements enforcing the boundary conditions. Additionally, the software enables us to define the permittivity values as a function of the local field power. Each of the presented results contains two simulation steps: a linear simulation where the optical parameters of the materials are considered linear or equivalently, assuming very low power operation, and a nonlinear simulation. We first performed linear simulations to obtain the linear reflection and transmission coefficients, and to obtain the resonance of the structure. In this simulation, we assume a unit cell of the metasurface with periodic boundary conditions on the left and right port, and excited the structure with periodic boundaries from top/bottom, while keeping the bottom/top port as receiving ports, as shown in Figure 11. Additionally, we added a nonuniform rectangle mesh to the unit cell, assuming a minimum element size of 38 nm inside the dielectric. A mesh view for the low-quality factor structure is shown in Figure 11 (left panel), and a distribution map of the mesh cell area is shown in Figure 11 (right panel). Notice that a similar mesh is used for the high-quality factor structure; however, it is important in this kind of structure to keep the mesh fixed in both the linear and nonlinear simulation, because any slight change in the mesh would significantly change the resonance location.

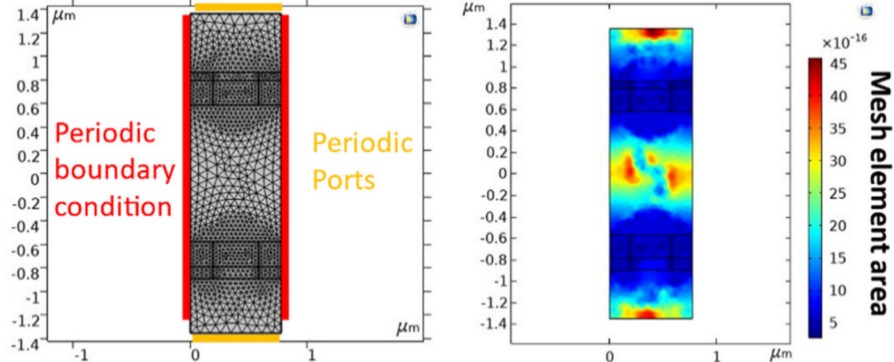

**Figure 11.** (**Left panel**) Setup of the frequency domain simulation showing that the structure is segmented in triangle areas with a maximum size of 38nm, and that it is excited by a periodic port either from the top or bottom, while keeping the periodic boundary condition on the left and right. (**Right panel**) Distribution map of the area of each mesh element; the scale bar unit is m$^2$.

After obtaining the linear response of the metasurface encoded in the reflection, transmission and energy, and determining the frequency of interest as presented in the main text, we updated the silicon permittivity using a user-defined function, such that the permittivity is written as $\epsilon = \epsilon_{linear} + \chi^{(3)}|E|^2$, where $\epsilon_{linear}$ is the permittivity used in the linear simulation, and the effect of nonlinearity is encoded in the second term of the permittivity. Notice that $|E|^2$ is not uniform inside the dielectric; therefore, our simulation takes into account all spatial inhomogeneity in the electric field. Then, we excited the unit cell with a plane wave of fixed frequency, and performed a parametric sweep over its power, in order to obtain the transmission from the top or bottom, respectively. Because the permittivity is a function of the induced field, the convergence of the solver is not always guaranteed. In order to overcome this problem, we increased the number of the iterations in the iterative solver to over 100 iterations in order to make sure that the solution will converge. In some situations, especially for the high Q structure, the convergence was not realized using the parameteric sweep, so we had to revert to manually updating the power and using the solution from the previous step as an initial condition for the next step. This is similar to following one curve on the bistability by either increasing or reducing the power. We repeat this step, but for excitation from the bottom to top in order to obtain the full transmission curves $T_{12}$ and $T_{21}$.

Overall, apart from the convergence and manually updating the initial condition, the solution time for both the linear and nonlinear problem takes less than five minutes, assuming 100 points of parametric sweep of power and wavelength. The simulation was performed on a personal laptop.

**Author Contributions:** Conceptualization, A.M., D.L.S. and A.A.; methodology, A.M., D.L.S. and A.A.; software, A.M.; validation, A.M., D.L.S.; formal analysis, A.M.; investigation, A.M.; resources, A.M.; data curation, A.M.; writing—original draft preparation, A.M.; writing—review and editing, A.M., A.A. and D.L.S.; visualization, A.M., D.L.S. and A.A.; supervision, A.A.; project administration, A.A.; funding acquisition, A.A. All authors have read and agreed to the published version of the manuscript.

**Funding:** This work was supported by the Air Force Office of Scientific Research MURI program, the National Science Foundation, and the Simons Foundation.

**Institutional Review Board Statement:** Not applicable.

**Informed Consent Statement:** Not applicable.

**Data Availability Statement:** The data in this paper are available by contacting the corresponding author.

**Conflicts of Interest:** The authors declare no conflict of interest.

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
