# Peer review of "Free-Space Nonreciprocal Transmission Based on Nonlinear Coupled Fano Metasurfaces"

_photonics, doi:10.3390/photonics8050139_

Round 1

Reviewer 1 Report

In the manuscript "Free-space nonreciprocal transmission based on nonlinear coupled Fano metasurfaces" by Ahmed Makkawy et al., the authors propose an all-dielectric flat metasurface made of coupled nonlinear Fano silicon resonant layers and realize large asymmetry in optical transmission at telecommunication frequencies. They start from the general design principle using a Fano resonance based on the guided mode resonance in double-layer silicon surface and include the Kerr nonlinearity of silicon, and present two implementations between two extremes 1) low quality factor resonators (prone to fabrication errors, but requiring large intensities in the range of GW/cm2) and (2) high quality factor resonators (more sensitive to fabrication errors but requiring much lower input power levels), respectively.

This work utilizes the emerging all-dielectric metasurface to realize free-space nonreciprocal transmission and achieve a transmission ratio >50 dB for oppositely propagating waves, an operational bandwidth exceeding 600 GHz, and insertion loss <0.04 dB, which should have merits in designing compact nonreciprocal components for optical communications. 

Overall, this is a good manuscript, technically sound, and showing quite interesting predictions. I don't have any objections and can recommend it for publication with some minor/optional suggestions.

Comments: 
1. The numerical model is not given in detail, for example, the simulations are conducted by code written by themself or by using the commercial software? The details of the numerical simulations, e.g. meshgrid setting, boundary condition, and running time, should be provided.

2. The authors propose the design principle using a Fano resonance based on all-dielectric metasurface. Some relevant articles can be included in the background introduction    Adv. Sci., 2019, 6(15): 1802119; Phys. Rev. Appl., 2019, 12(1): 014028; Phys. Rev. A, 2019, 100(6): 063803; ACS Photonics, 2020, 7(6): 1436-1443; Phys. Rev. B, 2020, 102(15): 155432.

3. The resolution of all the figures is low. The authors should replace all the figures with high resolution.

4. In Fig. 4, the specific value in the color bars should be clearly given, or the authors at least should ensure the color bars have already normalized throughout the work. 

5. The physical variables are required to be italic, while the subscripts should be normal. The authors should double check this throughout the manuscript. 

6. The English should be further improved in English/grammar/syntax which impede the smooth reading of the manuscript, especially in the introduction sections. 

Author Response

Reviewer 1

In the manuscript "Free-space nonreciprocal transmission based on nonlinear coupled Fano metasurfaces" by Ahmed Makkawy et al., the authors propose an all-dielectric flat metasurface made of coupled nonlinear Fano silicon resonant layers and realize large asymmetry in optical transmission at telecommunication frequencies. They start from the general design principle using a Fano resonance based on the guided mode resonance in double-layer silicon surface and include the Kerr nonlinearity of silicon, and present two implementations between two extremes 1) low quality factor resonators (prone to fabrication errors, but requiring large intensities in the range of GW/cm2) and (2) high quality factor resonators (more sensitive to fabrication errors but requiring much lower input power levels), respectively.

This work utilizes the emerging all-dielectric metasurface to realize free-space nonreciprocal transmission and achieve a transmission ratio >50 dB for oppositely propagating waves, an operational bandwidth exceeding 600 GHz, and insertion loss <0.04 dB, which should have merits in designing compact nonreciprocal components for optical communications. 

Overall, this is a good manuscript, technically sound, and showing quite interesting predictions. I don't have any objections and can recommend it for publication with some minor/optional suggestions.

We thank the reviewer for recommending the article for publication.

Comments: 
1. The numerical model is not given in detail, for example, the simulations are conducted by code written by themself or by using the commercial software? The details of the numerical simulations, e.g. meshgrid setting, boundary condition, and running time, should be provided.

We dedicated a separate sub-section in the methods and materials section to elaborate on the full numerical simulation details.

  1. The authors propose the design principle using a Fano resonance based on all-dielectric metasurface. Some relevant articles can be included in the background introduction    Adv. Sci., 2019, 6(15): 1802119; Phys. Rev. Appl., 2019, 12(1): 014028; Phys. Rev. A, 2019, 100(6): 063803; ACS Photonics, 2020, 7(6): 1436-1443; Phys. Rev. B, 2020, 102(15): 155432.

We thank the reviewer for recommending these relevant papers as examples for realizing Fano resonant metasurfaces and we added them in the paper. The references in the modified version are numbered [34]-[38].

  1. The resolution of all the figures is low. The authors should replace all the figures with high resolution.

We believe that this is due to a compression from the journal website. We now uploaded vector files for all figures in case they are not readable in the pdf.

  1. In Fig. 4, the specific value in the color bars should be clearly given, or the authors at least should ensure the color bars have already normalized throughout the work. 

We added the scale bar to the figure and made all of the figures with scale bar to be normalized to its maximum value so the scale bars are now from -1 to 1 for fields or from 0 to 1 for intensities.

5. The physical variables are required to be italic, while the subscripts should be normal. The authors should double check this throughout the manuscript. 

We updated this issue throughout the text.

  1. The English should be further improved in English/grammar/syntax which impede the smooth reading of the manuscript, especially in the introduction sections. 

We revised the current version and tried to update the manuscript where there may be deficiencies. Specifically for the introduction section we added more references related to nonreciprocal work using different techniques and elaborated on the differences between these works and our work.

Reviewer 2 Report

The paper proposes a nonreciprocal (NR) metasurface, where the nonreciprocity is achieved through nonlinearity of the structure.

First of all, for some reason, the quality of figures is extremely poor and unacceptable. I do not know if it is because of the compression caused by the journal platform or not. Please fix this issue.

The concept of isolation based on nonlinear Fano resonators is already published in another paper by the authors published in Nature Electronics. Hence, the current paper is only the spatial version of the previous structure.

Another important point is the lack of experimental demonstration. It is not clear why the authors have not carried out an experiment while the structure seems to be practically realizable.

Most importantly, the paper looks very weak because of the lack of an appropriate literature review, while some unnecessary references have been introduced. The paper proposes a nonreciprocal metasurface while recently proposed nonreciprocal metasurfaces based on alternative techniques have not been discussed. This includes nonreciprocal metasurfaces based on time modulation ([R1]-[R3]) and transistor-loaded cells ([R4]-[R7]).

The proposed metasurface should be characterized based on the recently proposed NR metasurfaces (e.g., [R1]-[R7]) and a fair comparison between this NR metasurface and other recently proposed NR metasurfaces (based on time modulation and transistor-loaded cells) should be presented in a table.

[R1] DOI: https://doi.org/10.1038/s41377-019-0225-z

[R2] DOI: https://doi.org/10.1103/PhysRevApplied.14.014027

[R3] DOI:https://doi.org/10.1103/PhysRevApplied.11.054054

[R4] DOI: https://doi.org/10.1073/pnas.1210923109

[R5] DOI: 10.1109/TAP.2017.2702712

[R6] DOI: https://doi.org/10.1038/s41598-021-86597-1

[R7] DOI: https://doi.org/10.1002/adom.201901285

Author Response

Reviewer 2

The paper proposes a nonreciprocal (NR) metasurface, where the nonreciprocity is achieved through nonlinearity of the structure.

First of all, for some reason, the quality of figures is extremely poor and unacceptable. I do not know if it is because of the compression caused by the journal platform or not. Please fix this issue.

We believe that this an issue due to compression from the journal website. We now uploaded vector files for all figures in case they are not readable in the pdf.

The concept of isolation based on nonlinear Fano resonators is already published in another paper by the authors published in Nature Electronics. Hence, the current paper is only the spatial version of the previous structure.

We agree with the reviewer that the current version is a spatial version of the electronic structure demonstrated before. We mention this in the manuscript “The device operation in Fig. 1a can be explained in analogy with the microwave circuits introduced in [33]”. But this is not the only contribution of this work. In fact, realizing controllable Fano resonators in electronic circuit is much easier compared to optical metasurfaces, since one can precisely choose the values of the capacitors, inductors a top of that the value of the nonlinear coefficient. However, in optical metasurface to realize two high quality factor resonances precisely chosen to have unitary power transmission when the incident power increased to certain level is a complicated idea, and we presented the solution for fully controlling the required power to enable bistability as shown in Fig. 6 in the main text. We also believe that this is the first contribution for multilayer flat optical metasurface that can have adjustable Fano resonances in a predicted manner as illustrated in the manuscript, above of that, controlling the switching power of each layer of the metasurface through attaching it to glass substrate that can control the amount of filed penetrable in the silicon slab is demonstrated.  Therefore, the presented approach can be easily scaled to any frequency based on flat metasurfaces that is very important for any photonic integrated circuit application. In fact, the reviewer suggests 7 references in the next comment that should be added to manuscript two of them are almost identical  R5, R6.

Another important point is the lack of experimental demonstration. It is not clear why the authors have not carried out an experiment while the structure seems to be practically realizable.

Our group carried out experimental demonstrations for one-layer metasurface. For the current structure however the power level of 16.5 GW/cm^2 is not available in our lab. Also, for 1.5 MW/cm^2 metasurface we need special type of lasers that works in continuous mode operation to ensure plane wave propagation and excite this very high Q structure.

Most importantly, the paper looks very weak because of the lack of an appropriate literature review, while some unnecessary references have been introduced. The paper proposes a nonreciprocal metasurface while recently proposed nonreciprocal metasurfaces based on alternative techniques have not been discussed. This includes nonreciprocal metasurfaces based on time modulation ([R1]-[R3]) and transistor-loaded cells ([R4]-[R7]).

The proposed metasurface should be characterized based on the recently proposed NR metasurfaces (e.g., [R1]-[R7]) and a fair comparison between this NR metasurface and other recently proposed NR metasurfaces (based on time modulation and transistor-loaded cells) should be presented in a table.

[R1] DOI: https://doi.org/10.1038/s41377-019-0225-zX. Guo, Y. Ding, Y. Duan, and X. Ni, “Nonreciprocal metasurface with space-time phase modulation,” Light Sci. Appl., vol. 8, no. 1, p. 123, 2019.

[R2] DOI: https://doi.org/10.1103/PhysRevApplied.14.014027  “S. Taravati and G. V. Eleftheriades, “Full-duplex nonreciprocal beam steering by time-modulated phase-gradient metasurfaces,” Phys. Rev. Appl., vol. 14, no. 1, 2020.

[R3] DOI: https://doi.org/10.1103/PhysRevApplied.11.054054J. W. Zang, D. Correas-Serrano, J. T. S. Do, X. Liu, A. Alvarez-Melcon, and J. S. Gomez-Diaz, “Nonreciprocal wavefront engineering with time-modulated gradient metasurfaces,” Phys. Rev. Appl., vol. 11, no. 5, 2019.

[R4] DOI: https://doi.org/10.1073/pnas.1210923109Z. Wang et al., “Gyrotropic response in the absence of a bias field,” Proc. Natl. Acad. Sci. U. S. A., vol. 109, no. 33, pp. 13194–13197, 2012.”               

[R5] DOI: 10.1109/TAP.2017.2702712  “S. Taravati, B. A. Khan, S. Gupta, K. Achouri, and C. Caloz, “Nonreciprocal Nongyrotropic Magnetless Metasurface,” IEEE Trans. Antennas Propag., vol. 65, no. 7, pp. 3589–3597, 2017.

[R6] DOI: https://doi.org/10.1038/s41598-021-86597-1S. Taravati and G. V. Eleftheriades, “Programmable nonreciprocal meta-prism,” Sci. Rep., vol. 11, no. 1, p. 7377, 2021.

[R7] DOI: https://doi.org/10.1002/adom.201901285Q. Ma et al., “Controllable and programmable nonreciprocity based on detachable digital coding metasurface,” Adv. Opt. Mater., vol. 7, no. 24, p. 1901285, 2019.

We dedicated two paragraphs for addressing the reviewer comment, adding all the mentioned references and we made a table comparing 9 metasurface, 7 of them as mentioned by the reviewer and we added two additional metasurface for comparison. The references suggested by the reviewer are numbered [11],[12],[13], [16],[17],[18],[19] in the modified version.

Reviewer 3 Report

This is an interesting contribution, which is above the average of the work submitted to this journal. The theory looks correct to me, however, it was impossible to judge some of the results. The quality of the figures is so poor that is difficult to read the axes and letters. Could you improve the quality of the figures in the pdf? 

Author Response

Reviewer 3

This is an interesting contribution, which is above the average of the work submitted to this journal. The theory looks correct to me, however, it was impossible to judge some of the results. The quality of the figures is so poor that is difficult to read the axes and letters. Could you improve the quality of the figures in the pdf? 

We thank the reviewer for their comment. We believe that this is a journal issue due to compression. We now uploaded vector files for all figures in case they are not readable in the pdf.

Round 2

Reviewer 2 Report

I have no further comments. The paper has been improved, especially its introduction, where the most relevant recently proposed transmissive nonreciprocal metasurfaces are discussed and compared with.